# Double-layer geodesic and gradient-index lenses

Qiao Chen [1], Simon A. R. Horsley [2], Nelson J. G. Fonseca [3], Tomáš Tyc [4] & Oscar Quevedo–Teruel [1✉]

A double-layer lens consists of a first gradient-index/geodesic profile in an upper waveguide, partially surrounded by a mirror that reflects the wave into a lower guide where there is a second profile. Here, we derive a new family of rotational-symmetric inhomogeneous index profiles and equivalent geodesic lens shapes by solving an inverse problem of pre-specified focal points. We find an equivalence where single-layer lenses have a different functionality as double-layer lenses with the same profiles. As an example, we propose, manufacture, and experimentally validate a practical implementation of a geodesic double-layer lens that is engineered for a low-profile antenna with a compact footprint in the millimeter wave band. Its unique double-layer configuration allows for two-dimensional beam scanning using the same footprint as an extension of the presented design. These lenses may find applications in future wireless communication systems and sensing instruments in microwave, sub-terahertz, and optical domains.

[1] School of Electrical Engineering and Computer Science, KTH Royal Institute of Technology, 11428 Stockholm, Sweden. [2] Department of Physics and Astronomy, University of Exeter, Stocker Road, Exeter EX4 4QL, UK. [3] Antenna and Sub-Millimetre Waves Section, European Space Agency, 2200 AG, Noordwijk, The Netherlands. [4] Institute of Theoretical Physics and Astrophysics, Faculty of Science, Masaryk University, Kotlářská 2, 61137 Brno, Czech Republic. ✉email: oscarqt@kth.se

When a material's refractive index changes in space gradually, an electromagnetic (EM) wave may be approximately described using a collection of rays following curved trajectories. This simplified view of electro-magnetism is the theory of geometrical optics[1], and its application to lens design goes back to Maxwell's consideration of the 'fish eye' lens, where all rays follow circular paths[2]. Although there are now more detailed design theories, where the entire EM field can be controlled (e.g., transformation optics[3,4]), geometrical optics remains a fascinating and active area of research.

Interestingly, in geometrical optics, the scattering by a rotationally symmetric profile $n(r)$ can be inverted[5–7]. Rather than finding the ray paths through the index profile, the index profile can be found from the ray deflection angle as a function of angular momentum. Perhaps most well-known of the profiles found from this inversion is the Luneburg lens[8], a circular disc (2D) or sphere (3D), that focuses all incident plane waves to points on the rim or surface of the lens. Other notable profiles include the Eaton lens[9,10], which acts as a perfect ray retro-reflector; Miñano's 'invisible lens'[11], where all rays make complete loops, leaving the device as if no refraction had occurred; Gutman's modified Luneburg lens[12–14], where the focus is inside the lens; and the multi-focal lenses found in the work of Šarbort and Tyc[7].

Graded index devices have been widely implemented using effective-medium techniques[10,14–21]. For instance, all-dielectric solutions have been employed for lensing from the microwave[10,15] to terahertz[16,17] and optical bands[18,19], and they also prove viable for both lensing and cloaking in quasi-conformal transformation optics[18–20]. Such devices are usually realized by a dielectric slab (such as alumina or silicon) patterned with subwavelength structures like pillars or air holes. They feature flat profiles and demonstrate low dissipation losses at the infrared range and beyond. However, in millimeter-wave bands low-loss dielectrics can be hard to find. In cases where the wave is confined to a plane, an all-metal metasurface lens realized in a textured parallel waveguide proves more efficient[21]. An interesting alternative are geodesic lenses that support either a surface wave[22] or the transverse electromagnetic (TEM) wave in a doubly-curved parallel plate waveguide. Since the TEM wave is non-dispersive, the geodesic lens provides ultrawide bandwidth. Here the out-of-plane deformation of the waveguide modifies the path length between any two points, equivalent to the change in optical path length due to a spatially varying index[23], $\int n dl$. The aforementioned inversion procedure can be adapted to designing geodesic lens shape from its functionality[6,24–28]. Such implementation can achieve a higher equivalent index than the effective-medium approaches that are limited by the available materials and geometrical parameters. For instance, sharp tips in the lens shape are equivalent to points of infinite refractive index (analogous to the transmutation of singularities with anisotropic media[29,30]), making it possible to realize a wider range of geodesic lenses than graded index ones[31]. Recent work has also shown that the theory of geodesic lenses can be used to circumvent some of the problems inherent in conformal transformation optics[32,33]. Additionally, the shielded structure of a geodesic lens permits multi-layer configurations that are not possible in an all-dielectric one due to evanescent interactions.

In this work, we explore a generalization of a recent problem addressed in lens design (schematic given in Fig. 1), where wave propagation takes place in two (upper and lower) parallel plate waveguides, containing two different inhomogeneous refractive index profiles and connected by a mirror or reflector over part of their common circumference. The combination of the reflection and the two inhomogeneous profiles serves to e.g., convert an incident plane wave in the first waveguide, to a focused spot in the

second. We call such devices 'double-layer' lenses. This idea of double-layer lenses was inspired by the work of Rotman[34] where the wave between two layers is transferred by a 'curved conducting back wall' at the lens rim, often referred to as a 'pillbox' antenna because of its shape. Thanks to the rotational symmetry, that solution provides a wider scanning range in comparison to parabolic reflectors (or a parabolic pillbox[35]) and other line source antennas, which made it an attractive solution for the early microwave radar systems developed in the 1950s. However, this geometry is limited by spherical aberrations. Combining the double-layer pillbox antenna described by Rotman and the Luneburg lens concept, a pillbox antenna with no aberrations (in principle) was recently proposed and referred to as a 'reflective Luneburg lens'[36].

In this paper, a generalization of this particular case is introduced, which defines a new family of inhomogeneous lenses with possible applications in the microwave, terahertz, and optical domains. The interest for quasi-optical parallel plate waveguide solutions has grown over recent years for use in terrestrial and non-terrestrial systems[14,21,27,35–37], making the more general solution presented here a timely and promising development. The proposed lens problem uses a Luneburg-like inversion of the ray propagation, but with the addition of a mirror at the edge of the profile (as in[34]). Before encountering the mirror the ray follows a curved trajectory in the upper inhomogeneous profile waveguide. It is then reflected into a lower waveguide where the ray follows a second curved trajectory in a different inhomogeneous profile. For the development reported here, it is assumed that the thickness of the waveguides is much smaller than the wavelength and that aberrations introduced by the 'curved conducting back wall' are negligible. We develop a general formula for the index profile. In the special case where the lower waveguide contains a homogeneous refractive index (or is perhaps a dielectric slab[35,38], or equivalent periodic surface[36,39] to generate leaky-wave radiation) we find a remarkable relation between double-layer and single-layer lenses. For instance, the generalized Maxwell fish eye lens profile employed in a double-layer lens provides the same functionality as the Eaton lens profile in the single-layer setting. Based on the double-layer generalized Maxwell fisheye lens, we design, fabricate, and test a fully-metallic geodesic lens antenna at 26–32 GHz with low profile and compact footprint for beam-scanning applications, whose experimental results exhibit extremely low insertion losses and stable gain patterns up to an angular range of ±50°. We demonstrate that the profile height of double-layer geodesic lenses can be significantly lowered by truncating and modulating the initial ones while keeping their functionalities, a general technique also applicable to other lenses. Furthermore, the distinctive double-layer configuration geometrically divides the lens functionality into a beamforming layer and a radiating one. The latter can be straightforwardly extended for 2D scanning by exploiting its entire footprint area as the radiation aperture.

## Results

**Problem formulation.** In geometrical optics, we treat the gradient of the wave's optical path as a 'velocity' $\boldsymbol{v} = \nabla S$, and the eikonal equation $(\nabla S)^2 = \boldsymbol{v}^2 = n^2$ is equivalent to the conservation of energy in classical mechanics, $\frac{1}{2} m \boldsymbol{v}^2 = E - V(\boldsymbol{x})$. For a 2D rotationally symmetric refractive index profile the angular momentum of each ray is also conserved

$$L = r v_\phi = r^2 \frac{d\phi}{dt} = \partial S / \partial \phi = \text{const.} \qquad (1)$$

Imposing this constraint allows us to eliminate $\partial S / \partial \phi$ from the eikonal equation, finding the radial velocity as a function

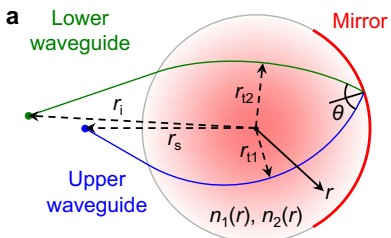
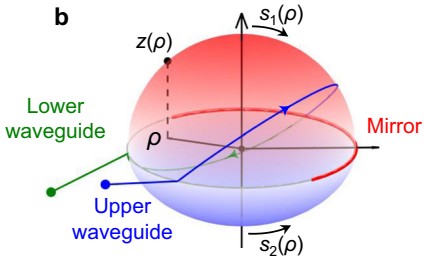

**Fig. 1 Schematic of the 2D double-layer lens problem. a** Graded index double-layer lens defined under polar coordinate system ($r, \phi$). Incident rays from a point source at $r = r_s$ in upper waveguide are first refracted by the profile $n_1(r)$ before encountering the mirror on the far side of the device. The mirror reflects the ray into a lower waveguide where it is refracted by the profile $n_2(r)$, focusing all rays to a point image at $r = r_i$. The thickness of the waveguides is neglected in first approximation for the purposes of this development, considering it is small compared to the wavelength. **b** Equivalent geodesic double-layer lens, where the refractive index is placed with an out of plane deformation $z(\rho)$ of the upper/lower waveguide described by the arc length $s_{1,2}(\rho)$ of the surface measured along the meridian from the axis of symmetry under a cylindrical coordinate system ($\rho, \phi, z$). In both cases, the background medium outside the lens has refractive indices $n_a$ and $n_b$ for the upper and lower waveguides, respectively.

of radius

$$\frac{dr}{dt} = \pm\sqrt{n^2(r) - \frac{L^2}{r^2}}. \quad (2)$$

Dividing Eq. (1) by (2) and integrating between radii $r_0$ and $r_1$, we find the associated change in the angular coordinate as a function of radius

$$\Delta\phi = \pm\int_{r_0}^{r_1} \frac{L}{r^2\sqrt{n^2(r) - \frac{L^2}{r^2}}}\, dr. \quad (3)$$

Given the conservation of angular momentum, the change in angle over the trajectory has a fixed sign. The sign is therefore chosen depending on whether the ray is traveling into or out from the origin. The sign changes at the turning points, where $n(r)$ $r = \pm L$. We now find the refractive index profile $n(r)$ for given source $r_s$, image $r_i$, and ray rotation angle $\Delta\phi$. The theory is a particular case of that given in Landau and Lifshitz[5] (section 12), and Šarbort and Tyc[6,7].

Consider the double-layer lens problem sketched in Fig. 1a. The rays originate from source point $r_s$, propagate through index $n_a$ up to the radius of the lens, which we choose as $r = 1$, without loss of generality. Within the lens, the ray propagates through the first index profile $n_1(r)$ from the outer radius to the turning point $r_{t1}$, and then returns to the outer radius. Meeting the outer radius for a second time it is reflected (as at a turning point) into a waveguide of background index $n_b \geq n_a$ and index profile $n_2(r)$, meeting at the final image point $r_i$. Note that the incidence direction is implicitly restricted due to the presence of the mirror at the lens rim, which breaks the rotational symmetry. Yet, wide-angle scanning properties are expected as in double-layer pillbox antennas[34].

Calculating the total change in angle $\Delta\phi$, for propagation between $r_s$ and $r_i$ as a sum of (3) for each segment of the trajectory, we find

$$\int_{r_{t1}}^{1} \frac{r^{-1}L\, dr}{\sqrt{n_1^2(r)r^2 - L^2}} + \int_{r_{t2}}^{1} \frac{r^{-1}L\, dr}{\sqrt{n_2^2(r)r^2 - L^2}}$$
$$= \frac{1}{2}\left[\Delta\phi - \int_{1}^{r_s} \frac{r^{-1}L\, dr}{\sqrt{n_a^2 r^2 - L^2}} - \int_{1}^{r_i} \frac{r^{-1}L\, dr}{\sqrt{n_b^2 r^2 - L^2}}\right] \equiv g(L) \quad (4)$$

where $g(L)$ is half the required turning angle of the double-layer lens. We carry out the constant index integrals in (4) in terms of inverse sin functions, each of which represents a contribution to

the total angle swept out by the ray outside the lens

$$\int_{r_a}^{r_b} \frac{r^{-1}L\, dr}{\sqrt{n_a^2 r^2 - L^2}} = \arcsin(L/n_a r_a) - \arcsin(L/n_a r_b). \quad (5)$$

Carrying out the integrals in this manner leaves us with a function $g(L)$ of the form

$$g(L) = \frac{1}{2}\left[\Delta\phi + \arcsin\left(\frac{L}{n_a r_s}\right) + \arcsin\left(\frac{L}{n_b r_i}\right) \right.$$
$$\left. - \arcsin\left(\frac{L}{n_a}\right) - \arcsin\left(\frac{L}{n_b}\right)\right]. \quad (6)$$

We now find the formula for the two refractive index profiles $n_{1,2}(r)$ such that the half turning angle equals (6). The integrals on left of Eq. (4) are closely related to Abel transforms[40], which can be inverted to find the refractive index $n(r)$. This inversion procedure is complicated in general because there are two integrals (one for each index profile) with different ranges of allowed angular momentum. There are however several special cases where we can simply invert the left-hand side of (5). In this work, we concentrate on two special cases: (i) where the two waveguides have equal background index $n_a = n_b$; and (ii) where the second waveguide contains a homogeneous index profile.

**Waveguides with equal background index $n_a = n_b$.** When the upper and lower waveguides have equal background index $n_a$, the maximum allowed angular momentum in both profiles is the same, equaling $n_a$. Multiplying both sides of (4) by $1/\sqrt{L^2 - \tilde{L}^2}$, where $\tilde{L}$ is some number between 0 and $n_a$, and integrating over $L$ from $\tilde{L}$ up to $n_a$ yields the turning points $r_{t1}$ and $r_{t2}$ as a function of the angular momentum variable $\tilde{L}$. We apply the integral identity

$$\int_{\tilde{L}}^{n_a} dL \int_{r_{t1}(\tilde{L})}^{1} \frac{r^{-1}L\, dr}{\sqrt{n_1^2(r)r^2 - L^2}} \frac{1}{\sqrt{L^2 - \tilde{L}^2}} = -\frac{\pi}{2}\log(r_{t1}(\tilde{L})) \quad (7)$$

to Eq. (4). Rearranging for the turning points in terms of the angular momentum we have

$$r_{t1}(\tilde{L})r_{t2}(\tilde{L}) = \exp\left(-\frac{2}{\pi}\int_{\tilde{L}}^{n_a} \frac{g(L)\, dL}{\sqrt{L^2 - \tilde{L}^2}}\right). \quad (8)$$

This formula shows that there is a freedom in the design of a double-layer lens, as we can only determine the *product* of the turning points. To make progress we write the turning point in the second index profile $r_{t2}(\tilde{L})$ as a function of $r_{t1}(\tilde{L})$. This function must be monotonic and such that it maps the interval $[0, 1]$ onto itself. As a particular case we choose the second

turning point to be a power of the first

$$r_{t2}(\tilde{L}) = [r_{t1}(\tilde{L})]^{\alpha} \qquad (9)$$

so that Eq. (8) becomes

$$r_{t1}(\tilde{L}) = \exp\left(-\frac{2}{\pi(1+\alpha)}\int_{\tilde{L}}^{n_a}\frac{g(L)\,dL}{\sqrt{L^2-\tilde{L}^2}}\right). \qquad (10)$$

This formula gives the radius where the refractive index satisfies $n(r_{t1})r_{t1} = \tilde{L}$. Even if the integral cannot be done analytically we can apply (10) to numerically generate a list of radii $r_{t1}$ and corresponding values of $n(r_{t1})r_{t1}$, from which we can compute the refractive index in the upper layer as a function of radius. To find the corresponding index profile in the lower layer we use Eq. (9) to write $n_2(r) = n_1(r^{1/\alpha})r^{1/\alpha-1}$. A further simplification of the general problem is to assume $n_1(r) = n_2(r)$ ($\alpha = 1$). In this case, the problem reduces to an ordinary inhomogeneous lens combined with a spherical mirror, with particular cases of interest reported in[11] and[41].

**Homogeneous index in lower waveguide**. When the lower waveguide contains a homogeneous index profile, $n_2(r) = n_b$, the second integral on the left of Eq. (4) can be evaluated using the integral identity in Eq. (5).

$$\int_{n_b/L}^{1}\frac{r^{-1}L}{\sqrt{n_b^2 r^2 - L^2}} = \frac{\pi}{2} - \arcsin(L/n_b) \qquad (11)$$

which leaves us with

$$\int_{r_{t1}}^{1}\frac{r^{-1}L\,dr}{\sqrt{n_1^2(r)r^2 - L^2}} = \tilde{g}(L) \qquad (12)$$

where $\tilde{g}(L)$ is the half turning angle of the upper index profile $n_1$

$$\tilde{g}(L) = \frac{1}{2}\left[\Delta\phi - \pi + \arcsin\left(\frac{L}{n_a r_s}\right) + \arcsin\left(\frac{L}{n_b r_i}\right)\right.$$
$$\left. + \arcsin\left(\frac{L}{n_b}\right) - \arcsin\left(\frac{L}{n_a}\right)\right] \qquad (13)$$

Applying the same Abel-type inversion procedure we used to obtain Eq. (8) from Eq. (4) we find the turning point in the upper index profile as a function of angular momentum

$$r_{t1}(\tilde{L}) = \exp\left(-\frac{2}{\pi}\int_{\tilde{L}}^{n_a}\frac{\tilde{g}(L)\,dL}{\sqrt{L^2-\tilde{L}^2}}\right) \qquad (14)$$

This formula enables us to find the refractive index profile $n_1(r)$ in the upper waveguide, in exactly the same manner as described below Eq. (10). Due to obvious symmetries in the problem formulation, a similar result can be obtained for a homogeneous index in the upper waveguide.

**Geodesic lenses**. The above theory can also be applied to design double-layer geodesic lenses that are considered in a cylindrical coordinate system $(\rho, \phi, z)$. To find the geodesic lens shape that is equivalent to an inhomogeneous index profile, we equate the optical length element $dl$ in the inhomogeneous profile to the physical distance on a deformed surface of variable height $z(\rho)$

$$dl^2 = n^2(r)\left(dr^2 + r^2 d\phi^2\right) = \left(1 + \left(\frac{dz}{d\rho}\right)^2\right)d\rho^2 + \rho^2 d\phi^2 \quad (15)$$

where the radial coordinate is $r$ in the inhomogeneous profile, and $\rho$ on the shaped surface. This equivalence leads us to identify $\rho(r) = n(r)r$ and $d\log(r) = \sqrt{1 + (dz/d\rho)^2}\,d\log(\rho) = s'(\rho)\,d\log(\rho)$, where $s(\rho)$ is the length on the deformed surface from the origin to the radius $\rho$.

Considering the first case of equal background indices in the two waveguides, the integral Eq. (4) becomes

$$\int_L^R\frac{L\,s_1'(\rho)d\rho}{\rho\sqrt{\rho^2 - L^2}} + \int_L^R\frac{L\,s_2'(\rho)\,d\rho}{\rho\sqrt{\rho^2 - L^2}} = g(L) \qquad (16)$$

where $R = n_a$ is the maximum radius of the geodesic lens, and $s'(\rho) = \sqrt{1 + (dz_{1,2}/d\rho)^2}$ represents the radial differential $s'(\rho) = ds/d\rho$ of the distance along the upper and lower geodesic surface, respectively, defined with reference to the rotational axis. This equation can be inverted using the same integral transform as (7),

$$s_1'(\rho) + s_2'(\rho) = -\frac{2\rho}{\pi}\frac{\partial}{\partial\rho}\int_\rho^R\frac{g(L)\,dL}{\sqrt{L^2-\rho^2}}. \qquad (17)$$

Again there is a freedom in designing the upper and lower geodesic shapes, and we can, for instance, add any $\rho$ dependent function to $s_1'$, if the same function is also subtracted from $s'$. To constrain the problem we can write e.g., $s_2'(\rho) = f(\rho)s_1'(\rho)$, where $f(\rho)$ is a monotonic function. However, we do not consider this case further here.

For the second case of interest, where the index profile in the lower waveguide is homogeneous and the lens shape is flat, Eq. (12) is equivalent to

$$\int_L^R\frac{Ls_1'(\rho)\,d\rho}{\rho\sqrt{\rho^2 - L^2}} = \tilde{g}(L) \qquad (18)$$

which can again be inverted to give the differential of the distance on the surface

$$s_1'(\rho) = -\frac{2\rho}{\pi}\frac{\partial}{\partial\rho}\int_\rho^R\frac{\tilde{g}(L)\,dL}{\sqrt{L^2-\rho^2}}. \qquad (19)$$

Substituting Eq. (13) into this integral and following a similar procedure as in[6], we can write

$$s_1'(\rho) = A + \frac{B}{\sqrt{1 - (\rho/n_a)^2}} \qquad (20)$$

where

$$A = \frac{1}{2} - \frac{1}{\pi}\left(\arcsin\sqrt{\frac{n_a^2 - \rho^2}{n_a^2 r_s^2 - \rho^2}} + \arcsin\sqrt{\frac{n_a^2 - \rho^2}{n_b^2 r_i^2 - \rho^2}}\right.$$
$$\left. + \arcsin\sqrt{\frac{n_a^2 - \rho^2}{n_b^2 - \rho^2}}\right) \qquad (21)$$

$$B = M - \frac{1}{2} + \frac{1}{\pi}\left(\arcsin\frac{1}{r_s} + \arcsin\frac{n_a}{n_b r_i} + \arcsin\frac{n_a}{n_b}\right). \qquad (22)$$

We have written $\Delta\phi = (M + 1)\pi$ in Eq. (13), in analogy with the notation of[6]. In both cases (17) and (19), we can find the shape of the lens from the relation $s'(\rho) = \sqrt{1 + (dz/d\rho)^2}$, first deriving the slope of the geodesic lens at all radii in terms of the half turning angle, and then integrating this to get the surface height $z(\rho)$.

Although in general $z(\rho)$ cannot be analytically computed, the closed-form solution of $s_1(\rho)$ can be obtained for the case where the lower waveguide is flat and has a homogeneous refractive index. Integrating Eq. (20) over $\rho$, the general solution of $s_1(\rho)$ is explicitly

expressed as

$$s_1(\rho) = A\rho + n_a B \arcsin\frac{\rho}{n_a} - \frac{1}{\pi}\left[ n_a r_s \arcsin\left(\rho\sqrt{\frac{r_s^2 - 1}{n_a^2 r_s^2 - \rho^2}}\right)\right.$$
$$+ n_b r_i \arcsin\left(\frac{\rho}{n_a}\sqrt{\frac{n_b^2 r_i^2 - n_a^2}{n_b^2 r_i^2 - \rho^2}}\right) + n_b \arcsin\left(\frac{\rho}{n_a}\sqrt{\frac{n_b^2 - n_a^2}{n_b^2 - \rho^2}}\right)$$
$$\left. - \left(n_a\sqrt{r_s^2 - 1} + \sqrt{n_b^2 r_i^2 - n_a^2} + \sqrt{n_b^2 - n_a^2}\right) \arcsin\frac{\rho}{n_a}\right]$$
(23)

where $1 \le n_a \le n_b$, and $r_{s,i}$ is normalized to $R = n_a$. From Eq. (23), we find that the homogeneous medium provides a degree of freedom that enables some interesting functionalities. For instance, if $n_a < n_b$ the profile can be designed as a 'magnifying lens'[42] that enhances the power density in the lower waveguide filled by index $n_b$. Also, the image point can be placed inside the lens rim as long as $r_i \ge n_a/n_b$. Note that when $n_a = n_b = 1$ and $r_{s,i} = 1$ or $\infty$, Eq. (23) reduces to $s_1(\rho) = A\rho + B \arcsin \rho$, which is detailed in the following sections.

**Examples**. We now give some simple double-layer lenses with homogeneous lower waveguide. To ease the discussion we set the refractive indices of the upper and lower waveguides to unity $n_a = n_b = 1$ in all cases, and assume foci that are either at infinity, or on the edge of the lens ($r = 1$).

In the simplest case, we take both foci at infinity $r_s = r_i = \infty$, such that half the turning angle of the lens (13) takes the form

$$\tilde{g}(L) = \frac{1}{2}(\Delta\phi - \pi) = \frac{M\pi}{2}$$
(24)

which is independent of the angular momentum $L$.

Substituting this function into our expression for the turning point (14) we find the turning point $r_{t1}$ as a function of the angular momentum

$$r_{t1}(\tilde{L}) = \frac{\tilde{L}^M}{\left(1 + \sqrt{1 - \tilde{L}^2}\right)^M}$$
(25)

By carrying out the same integral in (19), we can also find the shape of the equivalent geodesic lens

$$s'(\rho) = \sqrt{1 + \left(\frac{dz}{d\rho}\right)^2} = \frac{M}{\sqrt{1 - \rho^2}}$$
(26)

which implies that $s(\rho) = M \arcsin(\rho)$. This result can also be directly obtained from Eqs. (20), (23) with $A = 0, B = M$ by applying $r_s = r_i = \infty$ to Eqs. (21), (22). Substituting $\tilde{L} = n(r_{t1})r_{t1}$ into (26) and solving for $n(r)$, and integrating (26) to find the surface height $z(\rho)$, we find the following index profiles and surface heights for double-layer lenses with foci at infinity

$$n(r) = \frac{2}{r\left(r^{\frac{1}{M}} + r^{-\frac{1}{M}}\right)} \quad z(\rho) = -\int_0^\rho d\tilde{\rho}\sqrt{\frac{(M^2 - 1) + \tilde{\rho}^2}{1 - \tilde{\rho}^2}}$$
(27)

Interestingly the index profile (2), and equivalent geodesic shape (27) resemble those of the generalized Maxwell fish eye lens[43]. Note that only for $M \ge 1$ is $z(\rho)$ a real-valued function of $\rho$, which can be realized as a geodesic lens. This condition also ensures that the index (27) is greater than unity. The generalized fish eye profile can therefore be used either as a ordinary, or double-layer lens, with different functionality. In particular, as a double-layer lens the Maxwell Fish Eye profile ($M = 1$) (as a geodesic lens, this is a hemisphere $z(\rho) = \sqrt{1 - \rho^2} - 1$) has foci at infinity, with the rays turning once around the origin, $\Delta\phi = 2\pi$. We denote this lens as a double-layer 'invisible' lens with the idea that the functionality is

'folded' in the case of the double-layer configuration (see Table 1 and Section 2.1 for our naming convention). This lens has the equivalent effect to an Eaton lens, but without the need of infinity or a sharp tip in the lens center. In Fig. 2 we show ray propagation for four of the profiles (2), the integration of the eikonal equation and plotting of the ray trajectories was carried out using the Python SciPy[44], Numpy[45], and Matplotlib[46] libraries.

If one focus lies on the rim of the lens $r_s = 1$, and the other at infinity $r_i = \infty$ we have a double-layer lens with a function of an ordinary Luneburg lens or beam divider, depending on the value of $\Delta\phi$. In this case, the half turning angle (13) takes the form

$$\tilde{g}(L) = \frac{1}{2}[M\pi + \arcsin(L)]$$
(28)

Inserting this expression in (14) and carrying out the integral over $L$, the corresponding expression for the turning point as a function of $\tilde{L}$ is

$$r_{t1}(\tilde{L}) = \frac{\tilde{L}^M}{\left(1 + \sqrt{1 - \tilde{L}^2}\right)^{M+1/2}}$$
(29)

Substituting $\tilde{L} = n(r_{t1})r_{t1}$ into (29) we obtain an implicit equation for the refractive index as a function of radius

$$n^{2\alpha-2}r^{2\beta-2} - 2n^{\alpha-2}r^{\beta-2} + 1 = 0$$
(30)

where $\alpha = M/(M + 1/2)$ and $\beta = (M - 1)/(M + 1/2)$. Examples of the numerical solution to this equation are shown in the density plots within Fig. (3). We can also find the shape of the equivalent geodesic lens $s(\rho) = (M + 1/2) \arcsin(\rho) - \rho/2$. Again, this can be treated as a special case of the general solution in Eq. (23), thus obtained by letting $r_s = 1, r_i = \infty$ in Eqs. (21), (22). In general, we must numerically evaluate the refractive index (30) and the surface height. However, in the special case of $M = 1$, the implicit Eq. (30) reduces to a quadratic equation $n^{-4/3} - n^{-2/3}/2 - r^2/2 = 0$ in $n^{-2/3}$, which has the solution (taking the root where $n = 1$ at $r = 1$)

$$n(r) = \frac{8}{\left[1 + \sqrt{1 + 8r^2}\right]^{\frac{3}{2}}} = \frac{\left[\sqrt{1 + 8r^2} - 1\right]^{\frac{3}{2}}}{(2r^2)^{\frac{3}{2}}}.$$
(31)

This index profile is in agreement with that given in[36], where they studied the special case of the 'reflective' (here 'double-layer') Luneburg lens. In Fig. 3 we give the ray trajectories in several example profiles including this double-layer Luneburg lens. In all cases where $M$ is integer, the rays are converted from the diverging point source in the upper waveguide, to a single collimated beam in the lower guide. Meanwhile, when $M$ is non–integer the point source is divided into two beams at a relative angle of $2M\pi$.

As a final example, we consider a class of index profiles where both foci lie on the rim of the lens $r_s = r_i = 1$. As a single-layer problem, the solution would be the generalized Maxwell Fish Eye lens. However as we have already seen, the generalized Maxwell Fish Eye acts as a double-layer lens with foci at infinity. In this case, the half turning angle equals

$$g(L) = \frac{1}{2}[M\pi + 2 \arcsin(L)]$$
(32)

which compared to (28), only differs by a factor of two in front of the arcsin function. We can therefore straightforwardly find the turning point and the surface shape as before

$$r_{t1}(\tilde{L}) = \frac{\tilde{L}^M}{\left(1 + \sqrt{1 - \tilde{L}^2}\right)^{M+1}}$$
(33)

which in the same manner as the previous case, yields the same implicit equation for the refractive index (30), but with the

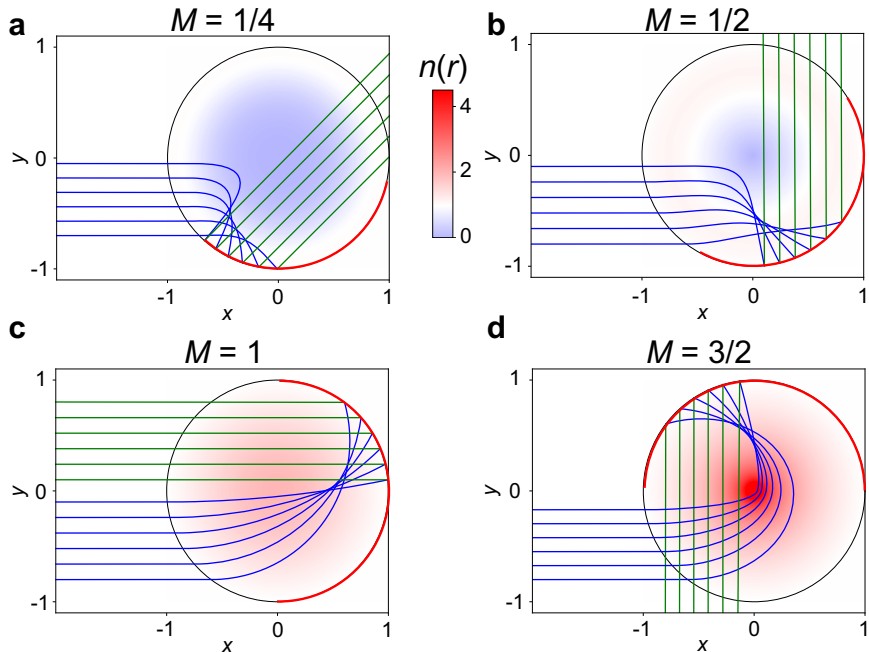

**Fig. 2 Double-layer lenses with foci at infinity.** The Generalized Maxwell Fish Eye profile can be used as a double-layer lens, with one homogeneous layer and foci at infinity. We show four examples, with ray turning angle $\Delta\phi$ of $\pi$ plus **a** $\pi/4$; **b** $\pi/2$; **c** $\pi$; and **d** $3\pi/2$. When $M$ is an odd integer, the lens provides the functionality of an ordinary Eaton lens; when $M$ is an even integer, it acts as an invisible lens where the rays travel in loops in the lens. Note that in Fig. 2 and the following figures the mirror coupling into lower waveguide is indicated as a red solid line, and the color of the rays (blue/green) indicates in which waveguide the propagation occurs (upper inhomogeneous index/lower homogeneous index).

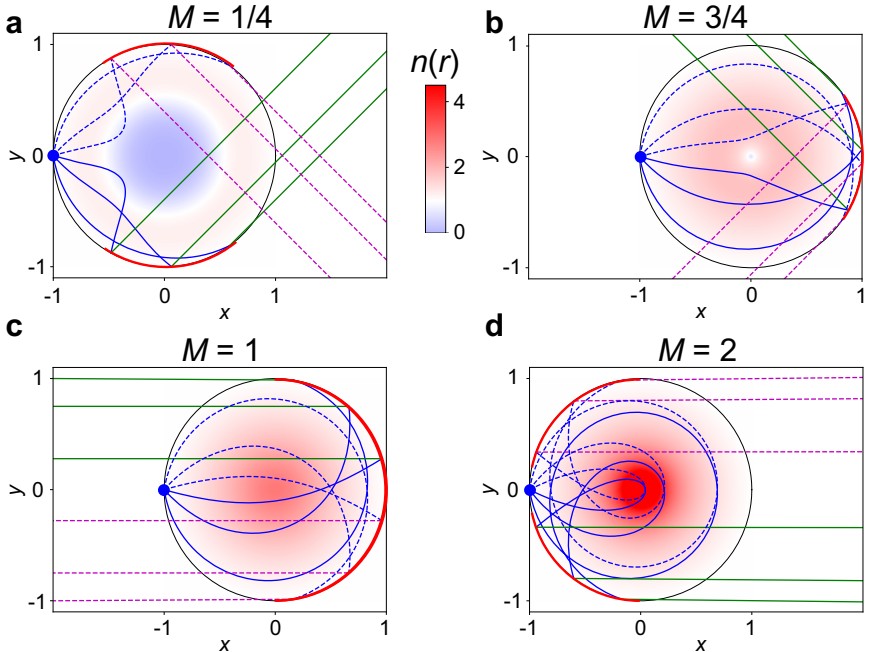

**Fig. 3 Double-layer lenses with one focus at $r = 1$.** From an initial point source at $r_s = 1$, $\phi = \pi$ (blue dot) we trace the ray motion through the profiles defined through the implicit Eq. (30) for different values of $M$. To aid visualization we plot the rays launched in the $+y$ direction as dashed lines, and those launched in the $-y$ direction as solid lines. We give four examples, where the turning angle $\Delta\phi$ is $\pi$ plus **a** $\pi/4$; **b** $3\pi/4$; **c** $\pi$; and **d** $2\pi$. When $M$ is integer, the double-layer lens has the functionality of a ordinary (for even numbers) and double-layer (for odd numbers) Luneburg lens, otherwise, it acts as a beam splitter fed by a point source.

modified powers $\alpha = M/(M+1)$ and $\beta = (M-1)/(M+1)$. Example refractive index profiles are shown as density plots in Fig. 4. The equivalent geodesic lens can also be found as $s(\rho) = (M+1)\arcsin(\rho) - \rho$, which is a special case of Eq. (23)

with $r_s = r_i = 1$ in Eqs. (21), (22). Again, the general index profiles and surface heights must be obtained numerically. However, for the particular case of $M = 2$ the implicit Eq. (30) reduces to a quadratic equation $n^{-4/3} - n^{-2/3}r^{1/3}/2 - r^{5/3}/2 = 0$ and we can

solve for the index profile analytically (taking the root where $n = 1$ at $r = 1$)

$$n(r) = \frac{8}{r^{1/2}\left(1 + \sqrt{1 + 8r}\right)^{3/2}}. \tag{34}$$

This profile has a square root singularity at $r = 0$ (the same strength of singularity as an ordinary Eaton lens), and is such that rays originating from a point source on the rim of the lens loop once around the origin before reflecting from the mirror to focus on the opposite side of the lens to the source. Propagation through this lens is shown in the final panel of Fig. 4. The examples of geodesic lenses are demonstrated in Supplementary Fig. 1.

**Comparison between double-layer and single-layer lenses.** We now compare the above special cases of Eq. (6) to the discussion given in ref. [6], where the equivalent general single-layer problem, referring here to the ordinary generalized Luneburg lens problem without mirror, is discussed for the case $n_a = n_b = 1$. In that work

the half turning angle is

$$g(L) = \frac{1}{2}\left[\Delta\phi + \arcsin\left(\frac{L}{r_s}\right) + \arcsin\left(\frac{L}{r_i}\right) - 2\arcsin(L)\right] \tag{35}$$

(Šarbort and Tyc [6]).

There is an extra factor of $\pi$ in our Eq. 13, i.e., $\Delta\phi \to \Delta\phi - \pi$, and the term $-2\arcsin(L)$ in (35) is absent from our expression. Both of these differences arise from the reversal of the ray at the mirror, which leads to an extra $\pi$ change in angle even in the absence of the refractive index profile.

Comparing Eq. (35) and Eq. (13) we can also understand the origin of the results. The generalized Fish Eye lens was observed to act as a retro-reflective lens with foci at infinity. Setting $r_s = r_i = 1$, Eq. (35) reduces to $g(L) = \Delta\phi/2$. This expression is identical to our form of $\tilde{g}(L)$ when $r_s = r_i = \infty$ in Eq. (6) ($n_a = n_b = 1$), with the understanding that $\Delta\phi \to \Delta\phi - \pi$.

To further aid comparison between single-layer and double-layer problems, we summarize our double-layer lens designs with one homogeneous layer in Table 1, where we also reproduce the

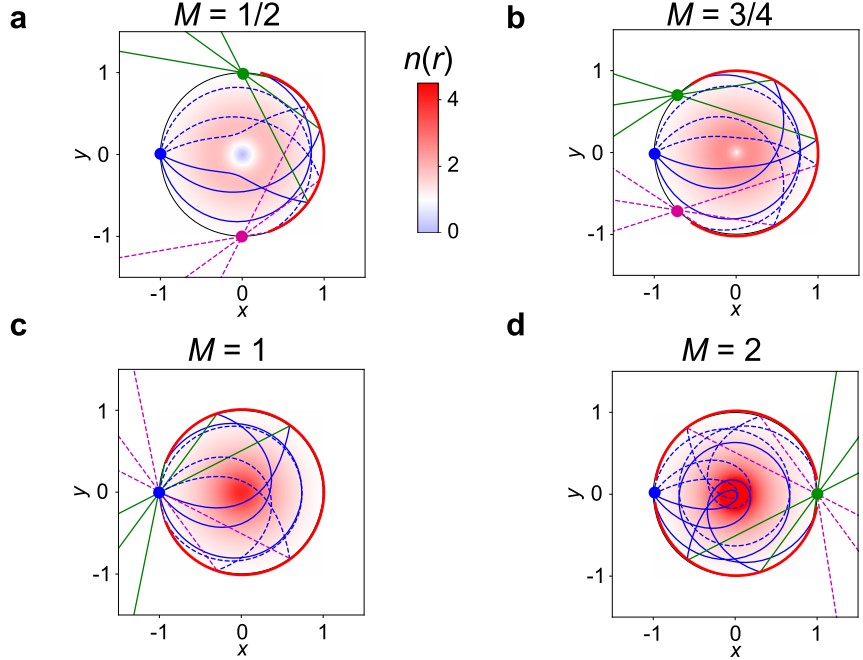

**Fig. 4 Double-layer lenses with both foci $r = 1$.** From an initial point source at $r_s = 1$, $\phi = \pi$ we trace the ray motion through the profiles defined through the implicit Eq. (30). To aid visualization we plot the rays launched in the $+y$ direction as dashed lines, and those launched in the $-y$ direction as solid lines. We give four examples, where the turning angle $\Delta\phi$ is $\pi$ plus **a** $\pi/2$; **b** $3\pi/4$; **c** $\pi$; and **d** $3\pi/2$. When $M$ is integer, the lens has the functionality of a ordinary (for even numbers) or double-layer (for odd numbers) Maxwell Fish Eye lens, while otherwise, it acts as a Generalized Fish Eye, where two foci appear on the rim.

**Table 1 Comparison between double-layer and single-layer lenses.**

| Single-layer Lens | $r_s$ | $r_i$ | $A$ | $B$ | $M$ | Double-layer lens | $r_s$ | $r_i$ | $A$ | $B$ | $M$ |
|---|---|---|---|---|---|---|---|---|---|---|---|
| MFE | 1 | 1 | 0 | 1 | 1 | DL-MFE | 1 | 1 | $-1$ | 2 | 1 |
| Generalized MFE | 1 | 1 | 0 | $M$ | $M$ | DL-generalized MFE | 1 | 1 | $-1$ | $M+1$ | $M$ |
| Luneburg | 1 | $\infty$ | 1/2 | 1/2 | 1 | DL-Luneburg | 1 | $\infty$ | $-1/2$ | 3/2 | 1 |
| Beam divider | 1 | $\infty$ | 1/2 | $M-1/2$ | $M$ | DL-beam divider | 1 | $\infty$ | $-1/2$ | $M+1/2$ | $M$ |
| Plane | $\infty$ | $\infty$ | 1 | 0 | 1 | | | | | | |
| 90° rot. | $\infty$ | $\infty$ | 1 | 1/2 | 3/2 | DL-90° rot. | $\infty$ | $\infty$ | 0 | 3/2 | 3/2 |
| Eaton | $\infty$ | $\infty$ | 1 | 1 | 2 | | | | | | |
| Invisible | $\infty$ | $\infty$ | 1 | 2 | 3 | DL-'invisible' | $\infty$ | $\infty$ | 0 | 1 | 1 |
| Beam divider | $\infty$ | $\infty$ | 1 | $M-1$ | $M$ | DL-beam divider | $\infty$ | $\infty$ | 0 | $M$ | $M$ |

On the left, we list the lenses of ref. [6], with $\Delta\phi = M\pi$, and geodesic lens shape $s(\rho) = A\rho + B\arcsin(\rho)$. On the right, we list our double-layer lenses where $\Delta\phi - \pi = M\pi$ (the general form of our expression for $s(\rho)$ is the same).

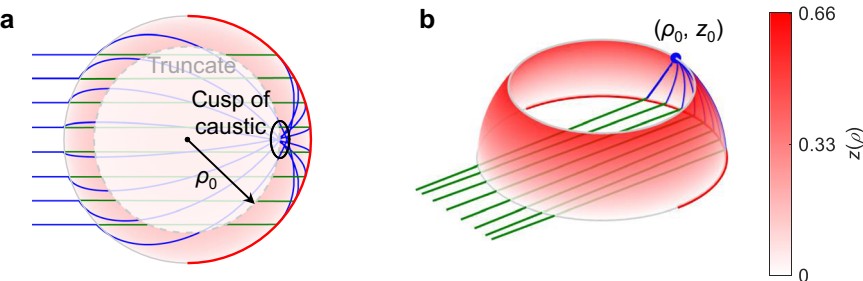

**Fig. 5 Truncation of lens. a** Ray trajectories and caustic (top view), **b** truncated double-layer `invisible' lens. The present example is with the dimensional parameters $\rho_0 = 0.75$ and $z_0 \approx 0.66$.

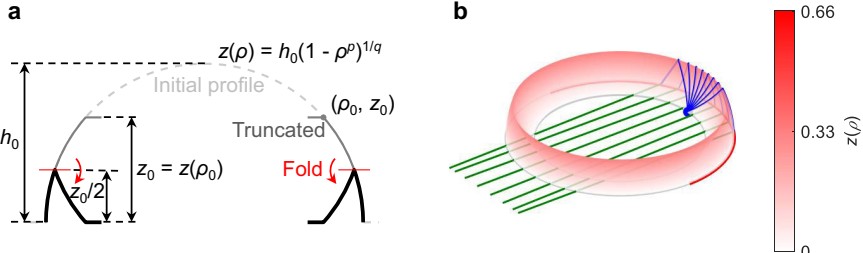

**Fig. 6 Folding of truncated lens. a** Evolution of lens profiles from the original version to a truncated and folded one, **b** 1-fold truncated double-layer `invisible' lens. The present example is with the dimensional parameters $h_0 = 1$, $p = q = 2$, $\rho_0 = 0.75$, and final profile height $z_0/2 \approx 0.33$.

analogous table of ref. [6]. We name the lenses found here using the values of $M$, $r_s$, and, $r_i$, i.e., the focal positions and the turning angle of the lens, accounting for the extra $\pi$ ray rotation due to the mirror. With this naming convention, the double-layer lenses provide the same functionality as a 'folded' lens of the same name. For example, the double-layer Luneburg lens has the source $r_s = 1$ and image $r_i = \infty$ on the same side of the lens, while an ordinary or single-layer Luneburg lens has the source and image on opposite sides of the lens. As a consequence, the outgoing rays are rotated by $\pi$ compared to the usual single-layer Luneburg lens. As a general rule we see that to obtain a double-layer lens with the same function as a single-layer one, we must peform the substitution $A \to A - 1$, and $B \to B + 1$.

**Geodesic double-layer lens antenna**. To link the previous findings to practical applications, we provide an example of an antenna design at 26–32 GHz based on the concept of the geodesic double-layer lens. In comparison to the conventional Luneburg lens antenna, our design features a more compact footprint beneficial for integration, while preserving its functionality and performance. The unique double-layer geometry makes possible the separation of the beamforming and the radiating layers by 'folding' the functionality of its conventional single-layer counterpart.

Although not demonstrated in this work, the standalone radiating layer permits the accommodation of an overlaying radiation aperture such as the one in[39] that produces 'pencil' beams in elevation. This way, the double-layer lens can exploit the area of its 2D footprint as the radiation aperture, fundamentally different to the Luneburg lens that radiates from its 1D periphery. Meanwhile, the beam scanning operation is the same as that of Luneburg lenses, i.e., by switching the feeding locations along a focal rim or mechanically steering the feed along that same focal rim.

Since the rays in a double-layer lens sweep over an additional angle $\pi$, a larger refractive index is needed. The implementation of such index typically requires the use of dielectrics that

inevitably introduce higher losses in the millimeter-wave band considered here. Alternatively, a fully-metallic structure can be implemented using the geodesic lens at an expense of a higher profile.

The height of the lens can be reduced by 'folding'[24], or more generally, 'modulating'[27] its profile curve $z(\rho)$. As long as the initial and the modified surfaces have the same variation of meridian length $s$ versus $\rho$ (i.e., $s(\rho)$ in both cases are the same), their rotational symmetry ensures that they have the same square of the variation, hence the same geodesics, namely, that they are equivalent to each other.

Another technique to decrease the height of the lens is to shorten the lengths of its geodesics while sustaining its functionality. This is achieved by cutting off the upper part of the surface above $z_0 = z(\rho_0)$, with $(\rho = \rho_0, \phi, z = z_0)$ being the contour coinciding with the cusp of the caustic formed by the rays on the lens surface. An illustration of this operation is exemplified in Fig. 5a by the double-layer 'invisible' lens presented in Fig. 2c (also Supplementary Fig. 1a). Then the rays launched from a point source at $(\rho_0, \phi, z_0)$ follow similar geodesics as the original ones in the remainder of the lens, as depicted in Fig. 5b. Since the feeding is placed at a focal region rather than a perfect focal point, the truncated lens inherently possesses aberrations causing uncollimated rays close to the aperture edges. As mentioned in Section 1, while the pillbox antenna[34] based on a planar circular reflector is subject to spherical aberrations, the double-layer lens produces a sharp image with no aberrations. Here, the truncated lens can be seen as a design trade-off between its focusing performance and profile height, providing fairly acceptable aberrations (as demonstrated later) with a moderate profile height. Generally, this truncating technique is applicable to reducing the maximum index/height of the generalized double-layer index/geodesic profiles listed in Table 1 as long as there is a cusped caustic in the lens. One example of its application to a graded-index lens based on a bed-of-nails structure is proposed in[36].

To further reduce the lens height, we combine both techniques, i.e., the lens truncation followed by a folding operation as

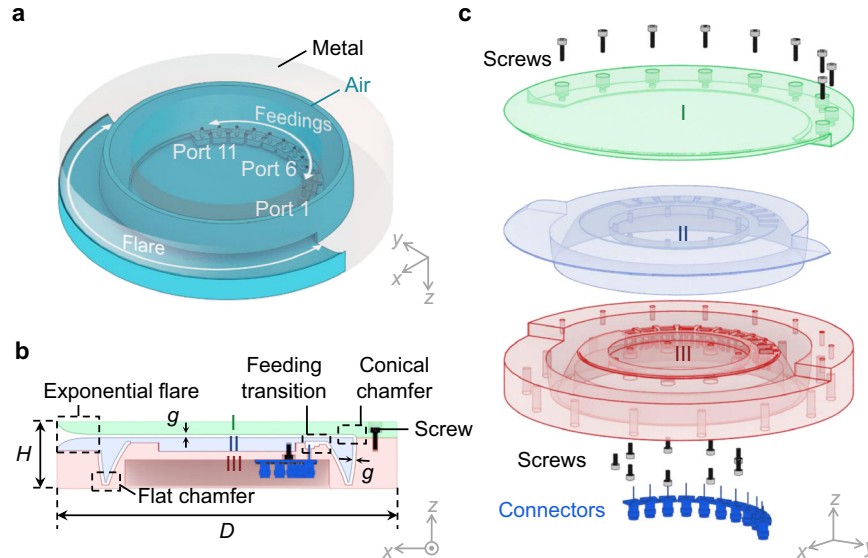

**Fig. 7 Implemented antenna structure. a** Perspective view of the functional hollow volume, **b** cross-sectional view, **c** isometric exploded view. The total diameter of the antenna is $D = 200$ mm, and the total height is $H = 40$ mm. The dimensional parameters of the lens profile are $h_0 = 1.23$, $p = 1.77$, $q = 2.28$, $\rho_0 = 0.78$, with lens radius $R = 75$ mm, and PPW height $g = 2$ mm.

illustrated by Fig. 6a. Here, the folding of the truncated surface is performed with respect to the symmetry plane $z = z_0/2$, leading to a 1-fold truncated double-layer (1F-TDL) lens with height $z_0/2$ and the feeding positioned at its inner periphery ($\rho = \rho_0$, $\phi$, $z = 0$). Apparently, an $N$-fold lens with height $z_0/2N$ can be obtained if a lower profile is required. Although the lens profile $z(\rho)$ does not usually have an analytical expression as remarked before, it can be well approximated (with residual errors of an order of magnitude $10^{-3}$) by a general superellipse (for $M \le 1$) following the same formulation as in[27]

$$z(\rho) = h_0(1 - \rho^p)^{1/q} \tag{36}$$

where $h_0$ is the lens height normalized to radius $R$, and $p$, $q$ characterize the profile shape. Eq. (36) is later used as a general formula to describe an optimized surface shape. For the 1F-TDL 'invisible' lens with $h_0 = 1$ exemplified here, the final profile height is roughly $h_0/3$, reduced by a factor of 4 in comparison to the double-layer Luneburg lens in Supplementary Fig. 1b with a height of 1.33 that provides a similar functionality, i.e., collimating rays launched from a point source at the lens rim. Since the geodesics remain the same before and after the folding, the 1F-TDL lens in Fig. 6b behaves exactly the same as the one in Fig. 5b. In addition to the height reduction, the 1F-TDL lens is expected to have lower energy losses in comparison to its complete version owing to the reduced optical paths in the lens.

As illustrated in Fig. 7, we implement the proposed 1F-TDL lens with a pair of curved conducting surfaces, referred to as a parallel plate waveguide (PPW) in microwave engineering, that confines the wave propagation along the geodesics. The functional hollow cavity formed by the PPW is highlighted in Fig. 7a. The PPW works in a TEM mode that is crucial for the lens to operate over a wide frequency bandwidth and, hence, the lens is frequency scalable. The lens is fed by 11 rectangular waveguide ports that are uniformly placed along its inner periphery, each one transitioned to a coaxial port for ease of measurement, as indicated in Fig. 7b. On the opposite side to the feeding, an exponential flare is added to the other layer for smoothly transforming the guided wave constrained in a narrow PPW into free-space radiation. Here, the PPW has a constant height $g = 2$ mm (smaller than quarter-wavelength)

everywhere in the lens except the two chamfers as annotated in Fig. 7b. As mentioned in Section 1, the conical chamfer at the outer rim of the lens acts as a reflector that collimates the rays in the top layer between plates I and II. To mitigate the reflections, a flat chamfer is introduced at the folding contour that creates a discontinuity in the surface. The reflections associated with these chamfers are found to be quite small and insensitive to the angle of incidence, which is evidenced by the low scattering coefficients measured at the testing ports as shown in Supplementary Fig. 2. To account for the impacts that the feeding, flare, and chamfering have on the radiation performance, the lens profile is optimized with the parameters defined in Eq. (36). Thanks to the good initial values obtained from the ray-tracing model, only a few iterations of phase-only optimization were performed by full-wave simulation, yielding the optimized dimensions $h_0 = 1.23$, $p = 1.77$, $q = 2.28$, $\rho_0 = 0.78$. Finally, the antenna structure occupies a cylindrical volume with diameter $D = 200$ mm and height $H = 40$ mm, sliced into three pieces that are stacked with screws as shown in Fig. 7c. The double-layer structure permits the feeding network to be deployed along the inner focal rim in a plane different from the radiating one. Therefore, it is enclosed by the overall volume of the antenna, offering a more compact mechanical footprint area in comparison to single-layer Luneburg lens antennas[21,27,47], where the feedings spread outwards from the outer rim in the same plane as the radiating flare. This compactness may be advantageous for a higher integration of the antenna system, enabling to fit electronic components connected to the ports directly in the volume formed by the antenna. Although a cylindrical symmetry is retained in this design, it is clear that its functional footprint can be still reduced by half by cutting it along the diameter in the $y$-axis and keeping only the reflector side as in[34].

In Fig. 8, we present the simulated surface current distributions in the antenna produced by two different ports. For the central beam in Fig. 8a, the uniform wavefront in the middle part of the aperture confirms a similar performance as the complete lens, while both edges are inefficiently illuminated due to the truncation that causes aberrations. These aberrations generate uncollimated rays from the edges of the reflector, also introducing slight interference visible in the illuminated region. Figure 8b demonstrates that the fields from an offset feeding behave similarly in the lens except that they are perturbed by the spillover and shadowing effects due to a

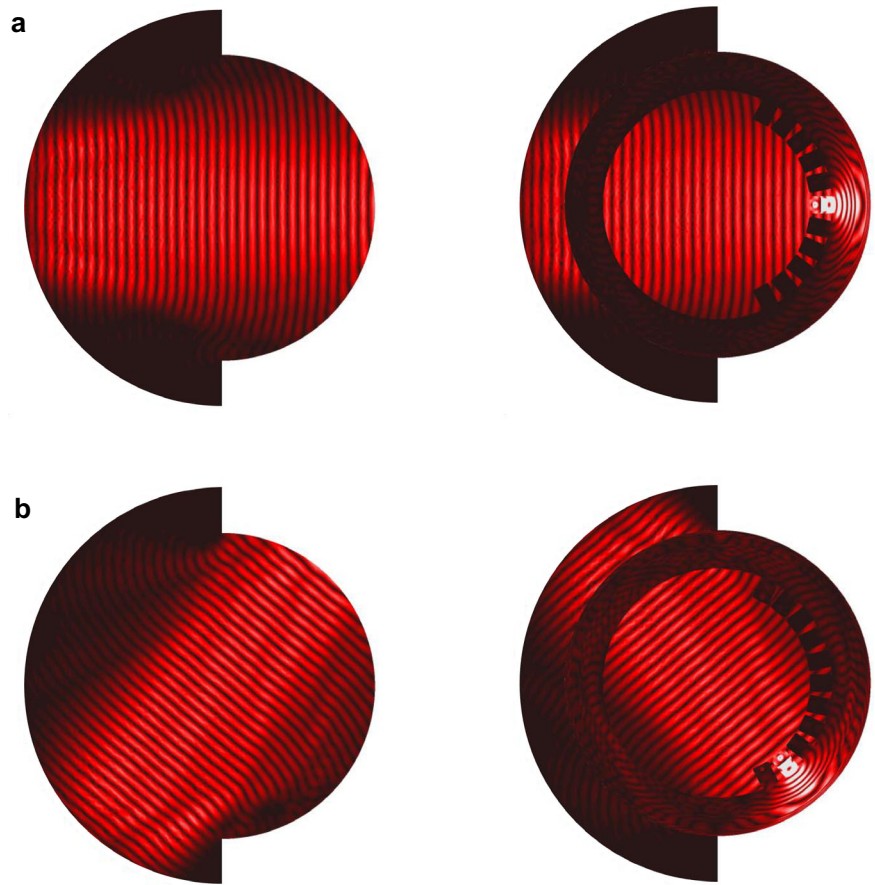

**Fig. 8 Simulated (ANSYS HFSS) surface current distribution at 30 GHz. a** Port 6, **b** Port 2.

broken rotational symmetry. Nevertheless, these imperfections are found to have minor impacts on the radiation performance, as demonstrated next.

A prototype is manufactured by aluminum milling, and measured in the far-field anechoic chamber with a spherical coordinate system defined in Fig. 9a. The measured realized gain patterns excited from three selected ports across the band are given in Fig. 9b. A stable radiation performance is observed up to $\Phi = \pm 50°$ for all sampled frequencies, with the gain, for instance at 30 GHz, varying from 20 to 21.1 dBi, as shown in Supplementary Table 1. Although not explicitly shown, this measured gain agrees well with the simulated (ANSYS HFSS) result varying within 20.5–21.4 dBi at the same frequency, exhibiting very low dissipation losses. The SLLs are all below −15 dB for the central beams in the band, and rise to roughly −13 dB at $\Phi = 50°$ for port 2. For the omitted last beam at an extreme angle $\Phi = 62.5°$ generated by port 1, the gain drops to 18.5 dBi at 30 GHz, with degraded SLLs of about −11 dB in the band. To fully demonstrate the radiation performance, 2D contours of the measured gain for beams excited by all ports are showcased in Fig. 9c. The antenna offers 'fan'-shaped beams in elevation that are directive and steerable in the azimuth plane, providing space diversity along one axis, of interest in terrestrial communication systems. These type of beams are also of interest for radar and radiometric airborne and spaceborne instruments. Furthermore, as mentioned in the beginning of this section, the second layer of the proposed antenna can be conveniently transformed into a planar radiating aperture that fully leverages its footprint area to realize 2-D pencil-beam switching with a low profile. This unique feature is highly demanded by broad applications such as 6G and satellite communications on the move (SOTM) user terminals.

## Discussion

We proposed the concept of the double-layer lens that is composed by a pair of rotational–symmetric profiles sharing a circumference partially bounded by a mirror. This mirror reflects the wave from one layer to a second layer which provides an extra degree of freedom for the lens design. We gave the general solution of index profiles and geodesic shapes for a family of such lenses by formulating a Luneburg-like inverse problem. Some special cases of particular interest were exemplified and closely examined, revealing the remarkable link between a known single–layer profile and its double-layer counterpart. This new family of lenses has a potential use for optical, terahertz, and microwave devices, including lens antennas and quasi-optical systems for ground and satellite communications, inter-satellite links, and radiometers and altimeters. As an example, we propose a varied version of a double-layer geodesic lens for prototyping a low-profile antenna at the $K_a$ up-link band of satellite communications. The experimental results demonstrate a comparable performance as the canonical Luneburg lens antenna with a more compact footprint area. Additionally, this footprint area can be later utilized as a radiation aperture to realize 2D scanning with highly-directive beams, an appealing asset for broader applications enabled by the distinctive double-layer geometry.

## Methods

The GO analysis is based on the zero-wavelength approximation of wave propagation, and it generally assumes that the medium properties and the wave field vary smoothly on the scale of the wavelength so the field behaves locally as a plane wave. The Maxwell equations, therefore, reduce to the eikonal equation that characterizes the wave by rays. Without loss of generality, let us consider a time-harmonic EM wave field in the source-free region of an isotropic medium. Its spatial component takes the

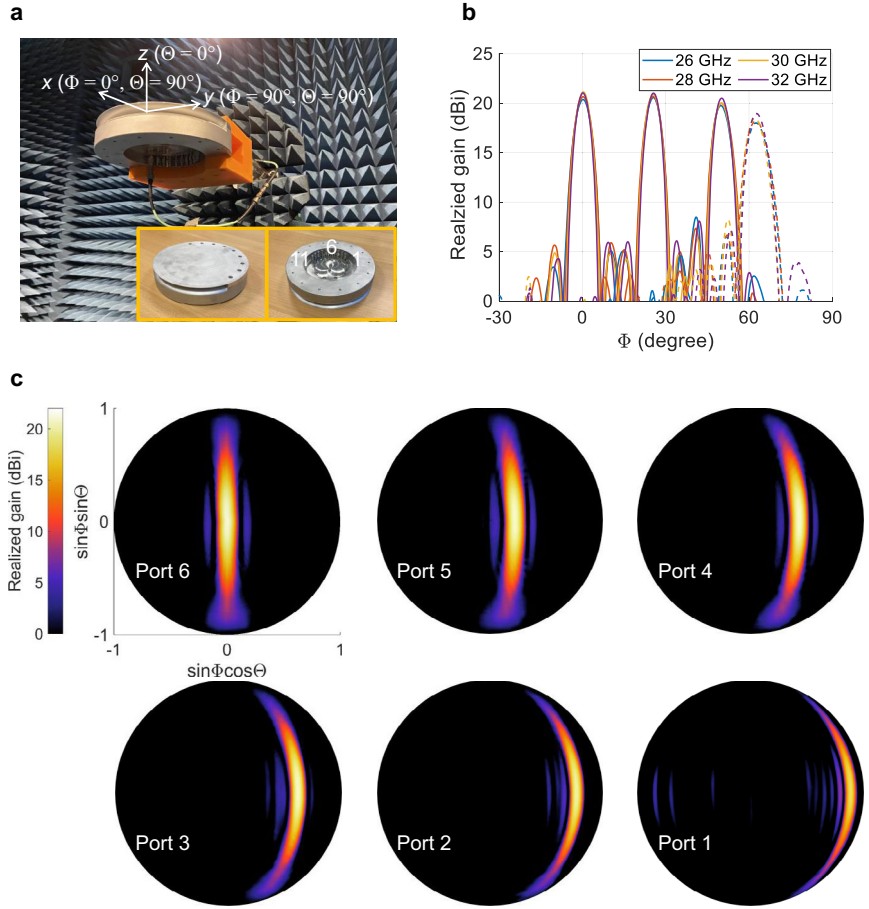

**Fig. 9 Prototype and measurement. a** Photo of the fabricated prototype and measurement setup in the far-field aechoic chamber. **b** Measured realized gain across the frequency band in the beamforming plane Θ = 90° for selected ports 6, 4, and 2. For brevity, only half of the angular space is presented due to symmetry. **c** Contour maps of the measured realized gain at 30 GHz over the angular range up to Φ = 62.5° for beams generated from Port 6-1, with the same scale.

form $U(\mathbf{r}) = u(\mathbf{r})\exp(-jk_0 S(\mathbf{r}))$, with $S$ being the optical path (surface $S$ = const. being the wavefront) and $u = \sum_{i=0}^{\infty} u_i/(jk_0)^i$ expressed as an asymptotic expansion in inverse powers of $jk_0$. Substituting $U$ into the Helmholtz equation $[\nabla^2 + k_0^2 n^2]U = 0$ yields the eikonal equation $|\nabla S|^2 = n^2$, where two conditions have been imposed: (i) the limit $k_0 \to \infty$ (equivalent to $\lambda_0 \to 0$) that gives $u = u_0$, and (ii) $\nabla^2 u_0/u_0 \ll k_0^2 n^2$ implying that the magnitude change of wave field is small in the order of the wavelength. The eikonal equation itself suggests a slow variation of the medium index and the wavefront. Accordingly, these conditions, though fulfilled in most applications with electrically large devices such as lenses and reflectors, limit the applicability of the ray-based model. First, since only the first term of the expansion is retained under the zero-wavelength condition, the ray-tracing analysis may be inadequate in problems such as diffraction and interference, where the higher-order terms or those with fractional powers dominate or lead to cumulative errors. The diffraction effects become more prominent when the device is not significantly larger than the wavelength. Second, the singularities in a medium such as poles and discontinuities that induce abrupt changes of the wave field may lead to the breakdown of the model in those positions. The laws of refraction and reflection in GO usually remain valid when the discontinuity interface is smooth per wavelength, and they may lose force otherwise. To extend GO to these exceptional situations, more general approximations must be adopted, such as the Kirchhoff's diffraction model useful in evaluating far-field patterns, and the WKB method which can be used to determine accurate wave fields in our radially symmetric index profiles. The readers are referred to[1,31,48] for more information.

## Data availability
The data that support the findings of this study are available from the corresponding author upon reasonable request.

## Code availability
The codes used to generate results in this paper are available from the corresponding author upon reasonable request.

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

## Acknowledgements

This article is based upon work from COST Action SyMat CA18223, supported by COST (European Cooperation in Science and Technology), www.cost.eu.

## Author contributions

Q.C., S.A.R.H., N.J.G.F, and O.Q.-T. conceptualized the work. S.A.R.H., Q.C., and T.T. carried out the analytical modeling and analysis. Q.C. and S.A.R.H. performed the ray-tracing analysis. Q.C. and N.J.G.F. performed the full-wave simulations. Q.C. designed the antenna prototype and conducted the experiments. Q.C. wrote the first version of the manuscript. O.Q.-T. planned, coordinated, and supervised the work, in consultation with T.T. All authors reviewed the manuscript.

## Funding

## Competing interests

The authors declare no competing interests.
