## [Peer review file · Nature Communications]

REVIEWER COMMENTS

Reviewer #1 (Remarks to the Author):

Thank you for an interesting paper. My comments are as follows

- 1) In the first sentence please make it clear the ray model is an approximation to how the em wave travels through such media, though a valuable one.
- 2) From your equation (8) your proposed solution is only one approach to the design i.e the profiles satisfying (8) are not unique. Please explain why you chose equation (9) et al to progress to a solution
- 3) On the measurements made the figure 11 is difficult to read please provide either further commentary or table so one can more clearly see the gain changes. As far as I can see the change of gain with scan angle has a different behaviour at different frequencies: It appears the 28GHz trace the gain drops with scan angle but not in the same way at 28ghz Please comment on why this is
- 4) Overall I feel the paper is overlong especially section IV reducing the number of examples should be considered. Sections V and VI could also be shortened without losing the essence of the paper
- 5) Somewhere in the paper you should address the limitations of this theoretical approach by which I mean when does the GO approximation start to break down (limitations of wavelength or how fast refractive index can be changed for example)

Reviewer #2 (Remarks to the Author):

The authors present rigorous analysis of a scenario in which two curved geodesic waveguide lenses are linked together via a circumferential mirror, as well as a range of example cases and experimental validation of the concept. This work is both scientifically interesting and potentially useful in the field of microwave and mm-wave engineering. I am willing to endorse the manuscript for publication in Nature Communications provided the following points are adequately addressed.

1. Please quantify the amount of reflection from the mirror back into the originating parallel-plate waveguide. The authors have employed a chamfer in order to reduce bending loss, but its performance is not reported. Does reflection depend upon the angle of incidence?
2. It appears that the concavity of the mirror itself is of great assistance with the intended beamforming functionality of this curved parallel-plate waveguide structure. Could the authors not eschew the curve

altogether, and employ a pair of planar parallel-plate waveguides that are connected by a curved mirror? It is significantly easier to construct planar parallel-plate waveguides than curved, and the resultant device will be compact and flat-profile. In fact, this approach has previously been employed for in-guide beam collimation, to realize high-frequency antennas (IEEE Trans. Antennas Propag. 68(2), 672-682, 2020). The authors should therefore cite this work, and justify the advantage of the curved parallel-plate waveguides.

3. The title is a little bit confusing, as many conventional/classical optics (i.e. those that interface with free-space fields) employ more than one layer of material. In fact, multi-layer eyeglasses are a commercial consumer product (<https://icueyewear.com/pages/lens-technology>). The authors have previously referred to this manner of curved parallel-plate-waveguide optic as a “geodesic” lens, and stated that it employs “non-Euclidean transformation optics,” in the title of Ref. 20, for example. The title of this manuscript should include some of these key-words in order to distinguish the work, and clearly articulate its novelty and contribution.

4. The third paragraph of the introduction reads “...graded index devices can be problematic. The requisite low-loss dielectrics can be hard to find, or are too heavy for practical applications.” This statement has issues that must be addressed. Firstly, the question of physical weight is a poor precedent to avoid dielectrics, as the authors employ metals to construct their parallel-plate waveguide devices; metals are quite heavy as well! Secondly, the authors essentially disregard more than a decade of progress in the field of GRIN optics devised using effective-medium techniques, which is highly relevant to the manuscript in review. Air may be mixed into high-index materials such as alumina or intrinsic silicon in a quasi-periodic arrangement of subwavelength pitch. In fact, the authors do cite one example of a Luneburg lens of this sort [28], but it is miscategorized in the final paragraph of the Introduction as a “quasi-optical parallel plate waveguide solution.” In reality, the antenna is composed entirely of silicon, and so it does not use parallel-plate waveguides. Closely related all-dielectric Luneburg lenses have been demonstrated in the infrared (Opt. Express 20(2), 1706-1713, 2012) and microwave (Sensors 11(8), 2011 7982-7991) ranges, and Maxwell fisheye lenses have also proven viable (J. Opt. 13, 024010, 2011; APL Photonics 6, 096104 2021)). Furthermore, effective-medium approaches can support quasi-conformal transformation optics (Nat. Mater. 8, 568–571 2009).

The authors must provide proper citation of the aforementioned all-dielectric alternative to their curved parallel-plate waveguides, and contrast the two approaches. There are several key tradeoffs to consider:

- Metals are quite lossy at high frequencies, and hence are not viable for the infrared range and above.
- The parallel-plate waveguide supports a TEM-wave, and hence is dispersionless and infinite-bandwidth.
- The dielectrics are entirely flat-profile, and do not require perturbation in the z-direction.

- In principle, the parallel-plate waveguide approach has no upper-bound upon the achievable range of effective index, as the z-span can be increased indefinitely. In contrast, The achievable range of effective index is limited by the constituent materials.

- The all-dielectric solution is unshielded, and so it cannot be made multi-layer due to evanescent interaction.

5. Is this concept strictly limited to two layers? I am interested in the possibility of a zig-zag profile in which a series of curved lenses are cascaded, giving the overall appearance of an accordion. This would necessarily entail reflection from a second mirror that is placed within the circumference of the parallel-plate waveguide structure. This manner of "spatial folding" could lead to drastic compression of the beamforming structure, leading to highly compact collimators and other such hand-held broadband microwave optics. Please provide some comments on the viability and benefit of this prospect.

6. The illustration in Fig. 7 suggests that only a small segment of the circle is involved in the desired beamforming functionality, and following reflection from the mirror at the circumference, the rays in the planar waveguide undergo no further manipulation. This suggests that most of the lens could simply be removed, yielding a more-compact device. Please comment on this possibility.

Reviewer #3 (Remarks to the Author):

This is a very interesting, comprehensive paper that develops a family of new electromagnetic devices all the way from ideas of Hamiltonian ray dynamics to tomographic design and finally a practical demonstration in the microwave domain. These new devices consist of two layers of different functionality. One is essentially responsible for the coupling of the radiation in and out of the device, while the other layer performs the desired function. The two layer are connected through a curved mirror: the radiation is reflected from the "upperworld" to the "underworld" where it is processed, and then leaves through the same mirror to the "upperworld" again where it couples out. In practical terms, this design concept of two different devices in one saves space and may allow for a wider functionality. Intellectually, the new concept follow a train of ideas starting from Hamilton's optics, Maxwell's graded index lenses in the early to mid 19th century, Luneburg's highly original lectures on mathematical optics in the 1950s to transformation optics - all in the best tradition of optics, which makes the paper both timeless and timely. I do not have much to add to this paper; it is very well written and easy to follow. Therefore I do recommend publication as is.

Responses to Reviewer #1

Reviewer #1, General comment: Thank you for an interesting paper. My comments are as follows

Authors' response: We would like to thank the reviewer for the generally positive feedback on our work.

Reviewer #1, Concern #1: In the first sentence please make it clear the ray model is an approximation to how the em wave travels through such media, though a valuable one.

Authors' response: Thank you for this comment.

Authors' action: We have revised the first sentence and now it reads:

“When a material’s refractive index changes in space gradually, an electromagnetic (EM) wave may be approximately described using a collection of rays following curved trajectories. ”

Reviewer #1, Concern #2: From your equation(8) your proposed solution is only one approach to the design i.e the profiles satisfying (8) are not unique. Please explain why you chose equation (9) et al to progress to a solution

Authors' response: In this work, our main focus is the case in Eq. (14) with a homogeneous layer. The case in Eq. (8) is also considered for the completeness and generality of the solution, and Eq. (9) is chosen for the brevity and the consistency of demonstration.

- Brevity. Since Eq. (8) reveals that the angular momentum is related to the product of two turning points, it is straightforward to choose one to be the power of the other such that Eq. (8) is transformed into a very simple form in Eq. (10).
 - Consistency. By choosing (9), (10) takes a similar form as (14) and, hence, it can be treated as a more general case of (14) taking $\alpha \neq 0$ and $\tilde{g}(L) \rightarrow g(L)$. The solving of (10) can then follow the same procedure as for (14) that is discussed throughout the manuscript.
-

Reviewer #1, Concern #3: On the measurements made the figure 11 is difficult to read please provide either further commentary or table so one can more clearly see the gain changes. As far as I can see the change of gain with scan angle has a different behaviour at different frequencies: It appears the 28GHz trace the gain drops with scan angle but not in the same way at 28ghz Please comment on why this is

Authors' response: Following the reviewer’s suggestion, we provide in Supplementary Information the measured gain data in Supplementary Table 1 (also reported below). It is found that the variation of the measured gain from Port 6 to Port 3 is mostly within ± 0.1 dB. Such magnitude is within the typical tolerance associated with manufacturing imperfections and measurement set-up errors such as misalignment, calibration, etc. Nevertheless, we can see that at all frequencies, the gain remains roughly stable from Port 6 to Port 3 and starts to drop at Port 2 due to shadowing effects. Meanwhile, the gain for each port increases with frequency as expected.

Authors' action: To make the manuscript compact, we provide Supplementary Table 1 in Supplementary Information. In the main text we refer the readers to this table when speaking of the gain and now it reads:

Supplementary Table 1: Measured realized gain of the prototype.

	Port 6	Port 5	Port 4	Port 3	Port 2	Port 1
26 GHz	20.4	20.4	20.6	20.5	19.8	18.0
28 GHz	20.7	20.7	20.7	20.6	20.0	18.2
30 GHz	21.1	20.8	20.9	20.7	20.0	18.5
32 GHz	21.0	21.0	21.0	21.1	20.5	19.0

“A stable radiation performance is observed up to $\Phi = \pm 50^\circ$ for all sampled frequencies, with the gain, for instance at 30 GHz, varying from 20 to 21.1 dBi, as shown in Supplementary Table 1.”

Reviewer #1, Concern #4: Overall I feel the paper is overlong especially section IV reducing the number of examples should be considered. Sections V and VI could also be shortened without losing the essence of the paper

Authors' response: Following the reviewer's comments, we have made several modifications at places (stated below) to shorten the manuscript partially, but we feel that further reduction would affect the clarity and flow of the paper. This paper introduces a general concept and we believe having an exhaustive discussion, as far as practical, of possible implementations is important to enhance the impact of this contribution, as we expect a number of variants of the design to find practical application.

Authors' action:

- We omit Fig. 5 where the examples of geodesic lenses are demonstrated. Now we put this figure in Supplementary Information as Supplementary Figure 1, with a referring sentence in the main text:
“Some examples of geodesic lenses are demonstrated in Supplementary Figure 1.”
- We omit the two integral identities in Eqs. (25)(31) that were purely technical and not mandatory for a clear comprehension of the development.
- We omit the solutions of geodesic lenses given in terms of $s'(\rho)$ in Eqs. (33)(38), and keep only the resulting $s(\rho)$ stated in the text. To keep the flow, we have also rearranged the order of the accompanying text (highlighted in blue) between Eqs. (20) and (21) about the general solution of geodesic lenses.
- The two quadratic equations displayed in Eqs. (34)(39) are now inserted in the text as inline formulas to save space.
- Figs. 10-12 are now combined as Fig. 9 to make the manuscript more compact.
- The abstract has been revised and shortened, complying with the editorial rule of maximum 150 words.

Reviewer #1, Concern #5: Somewhere in the paper you should address the limitations of this theoretical approach by which I mean when does the GO approximation start to break down (limitations of wavelength or how fast refractive index can be changed for example)

Authors' response: We have implemented your comment in the manuscript.

Authors' action: We have added a new section “IV. METHOD” where the applicability of the GO approximation is clarified. This new section reads:

“The GO analysis is based on the zero-wavelength approximation of wave propagation, and it generally assumes that the medium properties and the wave field vary smoothly on the scale of the wavelength so the field behaves locally as a plane wave. The Maxwell equations therefore reduce to the eikonal equation that characterizes the wave by rays. Without loss of generality, let us consider a time-harmonic EM wave field in the source-free region of an isotropic medium. Its spatial component takes the form $U(\mathbf{r}) = u(\mathbf{r}) \exp(-jk_0 S(\mathbf{r}))$, with S being the optical path (surface $S = \text{const.}$ being the wavefront) and $u = \sum_{i=0}^{\infty} u_i / (jk_0)^i$ expressed as an asymptotic expansion in inverse powers of jk_0 . Substituting U into the Helmholtz equation $[\nabla^2 + k_0^2 n^2]U = 0$ yields the eikonal equation $|\nabla S|^2 = n^2$, where two conditions have been imposed: i) the limit $k_0 \rightarrow \infty$ (equivalent to $\lambda_0 \rightarrow 0$) that gives $u = u_0$, and ii) $\nabla^2 u_0 / u_0 \ll k_0^2 n^2$ implying that the magnitude change of wave field is small in the order of the wavelength. The eikonal equation itself suggests a slow variation of the medium index and the wavefront. Accordingly, these conditions, though fulfilled in most applications with electrically large devices such as lenses and reflectors, limit the applicability of the ray-based model. First, since only the first term of the expansion is retained under the zero-wavelength condition, the ray-tracing analysis may be inadequate in problems such as diffraction and interference, where the higher-order terms or those with fractional powers dominate or lead to cumulative errors. The diffraction effects become more prominent when the device is not significantly larger than the wavelength. Second, the singularities in a medium such as poles and discontinuities that induce abrupt changes of the wave field may lead to the breakdown of the model in those positions. The laws of refraction and reflection in GO usually remain valid when the discontinuity interface is smooth per wavelength, and they may lose force otherwise. To extend GO to these exceptional situations, more general approximations must be adopted, such as the Kirchhoff’s diffraction model useful in evaluating far-field patterns and the WKB method which can be used to determine accurate wave fields in our radially symmetric index profiles. The readers are referred to [1,31,49] for more information.”

Responses to Reviewer #2

Reviewer #2, General comment: The authors present rigorous analysis of a scenario in which two curved geodesic waveguide lenses are linked together via a circumferential mirror, as well as a range of example cases and experimental validation of the concept. This work is both scientifically interesting and potentially useful in the field of microwave and mm-wave engineering. I am willing to endorse the manuscript for publication in Nature Communications provided the following points are adequately addressed.

Authors' response: Thank you for the overall positive evaluation of our work.

Reviewer #2, Concern #1: Please quantify the amount of reflection from the mirror back into the originating parallel-plate waveguide. The authors have employed a chamfer in order to reduce bending loss, but its performance is not reported. Does reflection depend upon the angle of incidence?

Authors' response: Thank you for this question. The main impact on performance is not due to the mirror itself but to the transition from the feeds to the geodesic lens, as most profiles require a discontinuity at the edge of the lens. The use of chamfers or toroidal bends was already suggested in early publications [24] and validated in recent practical implementations of geodesic lenses [27]. Reflection from the mirror and the flat chamfer will mostly translate into port-to-port coupling in multi-port designs and scattering losses in the case of a single-port design. With an adequate design of the mirror and the chamfer, these effects can be reduced. They can be quantified by examining the scattering parameters measured at the testing ports as shown in Supplementary Figure 2 (also attached below).

Supplementary Figure 2: Measured scattering parameters of the geodesic double-layer lens antenna. For brevity, only selected results are presented, and only the worst mutual-coupling coefficients are given. The coaxial-to-waveguide transition for testing works from 27 to 31 GHz.

We are interested in the band from 27 GHz to 31 GHz (highlighted by the solid black line) where the testing coaxial-to-waveguide transition is working properly. The reflection coefficients measured at different ports in the angular range 0-62.5° all exhibit similar patterns, with levels largely below -15 dB. The worst case is found in the extreme angle 62.5° at Port 1 due to the shadowing effects from a broken rotational symmetry. Meanwhile, the mutual couplings

measured between different ports are typically below -25 dB, with the worst case again, being the extreme angles at Port 1 and Port 11 due to the shadowing effects. This being said, when one port (e.g. Port 6 with negligible shadowing) is excited, the received power by itself and other ports are very low, meaning the reflections associated with the flat chamfer, the conical mirror, and the flare are quite small, and they are insensitive to the angle of incidence. This conclusion is also corroborated by the simulated surface current distribution in Fig. 9 (now Fig. 8) where no significant standing waves are observed.

Authors' action: Due to the maximum limit of 10 displayed items, we provide Supplementary Figure 2 in Supplementary Information. In the main text, we add a new sentence in Sec. II to refer the readers to this figure when speaking of the chamfers. It reads:

“The reflections associated with these chamfers are found to be quite small and insensitive to the angle of incidence, which is evidenced by the low scattering coefficients measured at the testing ports as shown in Supplementary Figure 2.”

Reviewer #2, Concern #2: It appears that the concavity of the mirror itself is of great assistance with the intended beamforming functionality of this curved parallel-plate waveguide structure. Could the authors not eschew the curve altogether, and employ a pair of planar parallel-plate waveguides that are connected by a curved mirror? It is significantly easier to construct planar parallel-plate waveguides than curved, and the resultant device will be compact and flat-profile. In fact, this approach has previously been employed for in-guide beam collimation, to realize high-frequency antennas (IEEE Trans. Antennas Propag. 68(2), 672-682, 2020). The authors should therefore cite this work, and justify the advantage of the curved parallel-plate waveguides.

Authors' response: The reviewer described two reference cases: a pair of planar parallel-plate waveguide connected at a common rim by i) a circular mirror, and ii) a parabolic mirror (in the paper given by the reviewer). They are usually referred to as “pillbox” structures, which are essentially double-layer *reflectors*. Therefore, the comparison between double-layer *lenses* and pillboxes is pretty much similar to the comparison between lenses and reflectors. Saying this, we can justify the advantage of double-layer lenses from the following aspects:

- **Aberrations.** The first case was systematically studied by Rotman in [26] (now [34]) where the circular mirror alone was proven capable of partially collimating the beam after reflection. So the reviewer is right that mirror itself plays an important role. However, this system is known to be subject to spherical aberrations due to the absence of a sharp focus. In fact, it was also concluded in [26] (now [34]) that combining the mirror with a geodesic shape significantly (though not perfectly) reduces the aberrations, which in effect is similar to our truncated implementation of the geodesic double-layer lens. However, that work conceptually differs from the main contribution of our work, i.e. we provide a solution with no aberrations in principle by combining the lens concept with the pillbox. One such example is illustrated in Fig.3c, where we show the double-layer Luneburg lens has a perfect focus in the infinity.
- **Scan losses.** As an alternative solution to mitigate aberrations, the circular mirror has revolved into better optimized shapes such as a parabola, i.e. the second case. Since the parabolic reflector has only one true focus, the antenna gain drops (and the pattern degrades) rapidly for the scanned angles offset from the focus. Although adding an auxiliary feeder or reflector can improve the scanning performance by introducing multiple foci, their total number cannot be more than a few (typically 2-5). However, thanks to the rotational symmetry, the proposed double-layer lens provides a continuous contour of foci or simply, a focal contour along the lens periphery. This enables large-angle scanning with stable gain pattern that cannot be achieved by pillbox-based solutions.
- **Profile height.** The double-layer lens can be implemented by either a gradient index or an out-of-plane shape. The former usually has a planar profile using e.g., a periodic surface such as the bed-of-nails structure in [27]

(now [36]). Alternatively, we propose the latter implementation that employs the geodesic lens with a simpler structure and lower losses at millimeter wave frequencies. We illustrate that the profile height can be notably reduced by applying either folding, modulation, or truncation, or a combination of any of those to the lens surface.

To summarize, many studies have been carried out ever since the concept was introduced trying to eliminate aberrations and scan losses of the pillbox-based solutions. As rightfully pointed out by the reviewer, this remains a very active field of research to this day, with new applications in the millimeter-waves. Nevertheless, the double-layer lens concept that combines a lens and a pillbox is the only solution so far that makes possible a wide-angle scanning without aberrations and the associated scan losses or pattern degradation. Such lens can be realized in a planar profile using periodic structures, or in a reduced profile using a folded/modulated and/or truncated geodesic lens. A design trade-off between the performance and the profile complexity must be made upon application, and the proposed design is a good trade-off for our target application. Thus, we believe this work to be a major advancement in this important field of research, opening opportunities for new designs with enhanced performance.

Here we took the circular pillbox in [26] (now [34]) as the main reference due to its rotational symmetry that is also adopted in our case. The motivation of using the double-layer lens rather than the pillbox was already justified at places:

- We stated the drawbacks of reflectors and pillboxes:
“Thanks to the rotational symmetry, that solution provides wider scanning range in comparison to parabolic reflectors and other line source antennas, However, this geometry is limited by spherical aberrations.”
- We stated the advantages of double-layer lens:
“Combining the double-layer pillbox antenna described by Rotman and the Luneburg lens concept, a pillbox antenna with no aberrations (in principle) was recently proposed and referred to as a “reflective Luneburg lens”[36]”
“Based on the double-layer generalized Maxwell fisheye lens, ... with low profile and compact footprint for beam-scanning applications, whose experimental results exhibit extremely low insertion losses and stable gain patterns up to an angular range of $\pm 50^\circ$.”
- We stated the trade-off between profile height and focusing performance:
“As mentioned in Sec. I, while the pillbox antenna [34] based on a planar circular reflector is subject to spherical aberrations, the double-layer lens produces a sharp image with no aberrations. Here, the truncated lens can be seen as a design trade-off between its focusing performance and profile height, providing fairly acceptable aberrations (as demonstrated later) with a moderate profile height.”

Authors’ action: Following the reviewer’s comment, we have stressed the motivation of using a double-layer lens by clearly linking the pillbox solution to reflectors. We have cited the mentioned reference paper in [35] (highlighted in the reference list) as an example of parabolic pillbox antennas, and it now reads:

- “Thanks to the rotational symmetry, that solution provides wider scanning range in comparison to parabolic reflectors (or a parabolic pillbox [35]) and other line source antennas,”

We have also cited it as an example of quasi-optical parallel-plate waveguide solutions, and an example of leaky-wave apertures in the introduction. They read respectively:

- “The interest for quasi-optical parallel plate waveguide solutions has grown over recent years for use in terrestrial and non-terrestrial systems [14, 21, 27, 35-37],”

- “In the special case where the lower waveguide contains a homogeneous refractive index (or is perhaps a dielectric slab [35,38], or equivalent periodic surface [36,39] to generate leaky-wave radiation)”

Reviewer #2, Concern #3: The title is a little bit confusing, as many conventional/classical optics (i.e. those that interface with free-space fields) employ more than one layer of material. In fact, multi-layer eyeglasses are a commercial consumer product (<https://icueyewear.com/pages/lens-technology>). The authors have previously referred to this manner of curved parallel-plate-waveguide optic as a “geodesic” lens, and stated that it employs “non-Euclidean transformation optics,” in the title of Ref. 20, for example. The title of this manuscript should include some of these key-words in order to distinguish the work, and clearly articulate its novelty and contribution.

Authors’ response: Thank you for this suggestion. The key novelty of the work lies on the “double-layer” configuration, which was already mentioned before. However, we do agree that including additional key-words in the title will certainly provide more visibility to this work.

Authors’ action: We have modified the title as:

“Double-Layer Geodesic and Gradient-Index Lenses”

Reviewer #2, Concern #4: The third paragraph of the introduction reads “...graded index devices can be problematic. The requisite low-loss dielectrics can be hard to find, or are too heavy for practical applications.” This statement has issues that must be addressed. Firstly, the question of physical weight is a poor precedent to avoid dielectrics, as the authors employ metals to construct their parallel-plate waveguide devices; metals are quite heavy as well! Secondly, the authors essentially disregard more than a decade of progress in the field of GRIN optics devised using effective-medium techniques, which is highly relevant to the manuscript in review. Air may be mixed into high-index materials such as alumina or intrinsic silicon in a quasi-periodic arrangement of subwavelength pitch. In fact, the authors do cite one example of a Luneburg lens of this sort [28], but it is miscategorized in the final paragraph of the Introduction as a “quasi-optical parallel plate waveguide solution.” In reality, the antenna is composed entirely of silicon, and so it does not use parallel-plate waveguides. Closely related all-dielectric Luneburg lenses have been demonstrated in the infrared (Opt. Express 20(2), 1706-1713, 2012) and microwave (Sensors 11(8), 2011 7982-7991) ranges, and Maxwell fisheye lenses have also proven viable (J. Opt. 13, 024010, 2011; APL Photonics 6, 096104 2021)). Furthermore, effective-medium approaches can support quasi-conformal transformation optics (Nat. Mater. 8, 568–571 2009).

The authors must provide proper citation of the aforementioned all-dielectric alternative to their curved parallel-plate waveguides, and contrast the two approaches. There are several key tradeoffs to consider:

- Metals are quite lossy at high frequencies, and hence are not viable for the infrared range and above.
- The parallel-plate waveguide supports a TEM-wave, and hence is dispersionless and infinite-bandwidth.
- The dielectrics are entirely flat-profile, and do not require perturbation in the z-direction.
- In principle, the parallel-plate waveguide approach has no upper-bound upon the achievable range of effective index, as the z-span can be increased indefinitely. In contrast, The achievable range of effective index is limited by the constituent materials.
- The all-dielectric solution is unshielded, and so it cannot be made multi-layer due to evanescent interaction.

Authors' response: Thank you for pointing this out. We have implemented your comments in the manuscript.

Authors' action:

- We have amended the improper citation of the [28] (now [16]).
- We have cited all mentioned reference papers of all-dielectric solutions in [15,17-20] highlighted in the reference list.
- We have revised the third paragraph of the introduction following the key points suggested by the reviewer. Now it reads:

“Graded index devices have been widely implemented using effective-medium techniques [10,14–21]. For instance, all-dielectric solutions have been employed for lensing from the microwave [10,15] to terahertz [16,17] and optical bands [18,19], and they also prove viable for both lensing and cloaking in quasi-conformal transformation optics [18-20]. Such devices are usually realized by a dielectric slab (such as alumina or silicon) patterned with subwavelength structures like pillars or air holes. They feature flat profiles and demonstrate low dissipation losses at the infrared range and beyond. However, in millimeter wave bands low-loss dielectrics can be hard to find. In cases where the wave is confined to a plane, an all-metal metasurface lens realized in a textured parallel waveguide proves more efficient [21]. An interesting alternative are geodesic lenses that support either a surface wave [22] or the transverse electromagnetic (TEM) wave in a doubly-curved parallel plate waveguide. Since the TEM wave is non-dispersive, the geodesic lens provides ultrawide bandwidth. Here the out-of-plane deformation of the waveguide modifies the path length between any two points, equivalent to the change in optical path length due to a spatially varying index [23], $\int n dl$. The aforementioned inversion procedure can be adapted to designing geodesic lens shape from its functionality [6, 24-28]. Such implementation can achieve a higher equivalent index than the effective-medium approaches that are limited by the available materials and geometrical parameters. For instance, sharp tips in the lens shape are equivalent to points of infinite refractive index (analogous to the transmutation of singularities with anisotropic media [29, 30], making it possible to realise a wider range of geodesic lenses than graded index ones [31]. Recent work has also shown that the theory of geodesic lenses can be used to circumvent some of the problems inherent in conformal transformation optics [32, 33]. Additionally, the shielded structure of a geodesic lens permits multi-layer configurations that are not possible in an all-dielectric one due to evanescent interactions.”

Reviewer #2, Concern #5: Is this concept strictly limited to two layers? I am interested in the possibility of a zig-zag profile in which a series of curved lenses are cascaded, giving the overall appearance of an accordion. This would necessarily entail reflection from a second mirror that is placed within the circumference of the parallel-plate waveguide structure. This manner of “spatial folding” could lead to drastic compression of the beamforming structure, leading to highly compact collimators and other such hand-held broadband microwave optics. Please provide some comments on the viability and benefit of this prospect.

Authors' response: This is an interesting idea. The reviewer has in mind a folding that takes place in the radial direction (instead of z direction), which would correspond to having multiple values of z for one value of r (in the paper we consider two). Conceptually this is possible as an extension of the proposed double-layer lens, and this could potentially relax the requirement for index or profile height, making the device more compact as the reviewer expected. One should note, however, that conservation of the angular momentum L of the ray with respect to the axis of the surface (z axis) would prevent some rays, namely those with a large L, getting close enough to the axis. These rays would then not be able to explore the entire surface, and in particular they would not be able to get to the inner mirror (whose radius must be less than unity to fit inside the outer, r=1 mirror). Therefore such a structure could work only for some range of rays but not all, which would be a disadvantage. This limitation is equivalent to

that of multi-reflector optics, and the problem may be better understood as cascading spillover losses will eventually reduce the efficiency of the overall beamforming design, the higher the number of optics the higher the losses. These configurations are generally preferred when no scanning is required. By analogy, it is anticipated the same will be true for the proposed multi-layer lens configuration. So one can anticipate that the concept proposed by the reviewer could be beneficial for applications requiring limited or no scanning. For this reason we do not consider such a type of structure, even though we admit it might have some interesting features.

Reviewer #2, Concern #6: The illustration in Fig. 7 suggests that only a small segment of the circle is involved in the desired beamforming functionality, and following reflection from the mirror at the circumference, the rays in the planar waveguide undergo no further manipulation. This suggests that most of the lens could simply be removed, yielding a more-compact device. Please comment on this possibility.

Authors' response: The reviewer is right that the footprint of the lens could have been reduced to half by keeping only the mirror side. This would lead to a design in a similar appearance of [26] (now [34]), which would radiate via a flat aperture. However, we decided to keep the rotational symmetry for a better visual link to the proposed lens concept. As a key advantage, when the lens is combined with an overlaying radiation aperture, its rotational symmetry permits 360° coverage in azimuth and a wider-angle scanning in elevation.

Authors' action: The possibility of further reducing the lens was already stated in Sec. V.B (now Sec. II before referring to Fig. 8). To increase the visibility of this statement, we have added here a reference to [34] and it now reads:

“Although a cylindrical symmetry is retained in this design, it is clear that its functional footprint can be still reduced by half by cutting it along the diameter in the y -axis and keeping only the reflector side as in [34].”

Responses to Reviewer #3

Reviewer #3, General comment: This is a very interesting, comprehensive paper that develops a family of new electromagnetic devices all the way from ideas of Hamiltonian ray dynamics to tomographic design and finally a practical demonstration in the microwave domain. These new devices consist of two layers of different functionality. One is essentially responsible for the coupling of the radiation in and out of the device, while the other layer performs the desired function. The two layer are connected through a curved mirror: the radiation is reflected from the "upperworld" to the "underworld" where it is processed, and then leaves through the same mirror to the "upperworld" again where it couples out. In practical terms, this design concept of two different devices in one saves space and may allow for a wider functionality. Intellectually, the new concept follow a train of ideas starting from Hamilton's optics, Maxwell's graded index lenses in the early to mid 19th century, Luneburg's highly original lectures on mathematical optics in the 1950s to transformation optics - all in the best tradition of optics, which makes the paper both timeless and timely. I do not have much to add to this paper; it is very well written and easy to follow. Therefore I do recommend publication as is.

Authors' response: Thank you for the great endorsement of our work. We are glad to read that the reviewer could see the key contribution of this work and how it fits in the long history of this very important field of research. We certainly hope readers will benefit from our manuscript and find it equally interesting.

REVIEWERS' COMMENTS

Reviewer #1 (Remarks to the Author):

Thank you for your revised paper. I feel the revised manuscript has captured well both my own points and those of other reviews. I think it is now a interesting and useful piece of work and am happy for it to be published

Reviewer #2 (Remarks to the Author):

The authors have adequately addressed all of my concerns. I recommend the present form of the manuscript for publication.

Responses to Reviewer #1

Reviewer #1, Final comment: Thank you for your revised paper. I feel the revised manuscript has captured well both my own points and those of other reviews. I think it is now a interesting and useful piece of work and am happy for it to be published.

Authors' response: We thank the reviewer for the effort and time devoted to reviewing our work. We are glad to have your positive evaluation of our work.

Responses to Reviewer #2

Reviewer #2, Final comment: The authors have adequately addressed all of my concerns. I recommend the present form of the manuscript for publication.

Authors' response: We thank the reviewer for the positive feedback of our work. We appreciate the reviewer's effort and time devoted to reviewing our work.